# Mechanical Behaviors of the Origami-Inspired Horseshoe-Shaped Solar Arrays

**DOI:** 10.3390/mi13050732

**Published:** 2022-05-02

**Authors:** Zhi Li, Chengguo Yu, Luqiao Qi, Shichao Xing, Yan Shi, Cunfa Gao

**Affiliations:** 1State Key Laboratory of Mechanics and Control of Mechanical Structures, Nanjing University of Aeronautics & Astronautics, Nanjing 210016, China; zhi.li@nuaa.edu.cn (Z.L.); lqqi@nuaa.edu.cn (L.Q.); cfgao@nuaa.edu.cn (C.G.); 2Xi’an Research Institute of High Technology, Xi’an 710025, China; yuchengguo@cardc.cn; 3China Aerodynamics Research and Development Center, Mianyang 621000, China; 4Key Laboratory of Construction Hydraulic Robots of Anhui Higher Education Institutes, Tongling University, Tongling 244061, China; xingsc@nuaa.edu.cn

**Keywords:** solar arrays, horseshoe-shaped structure, shooting method, finite deformation theory, experiment

## Abstract

The importance of flexibility has been widely noticed and concerned in the design and application of space solar arrays. Inspired by origami structures, we introduce an approach to realizing stretchable and bendable solar arrays via horseshoe-shaped substrate design. The structure has the ability to combine rigid solar cells and soft substrates skillfully, which can prevent damage during deformations. The finite deformation theory is adapted to find the analytic model of the horseshoe-shaped structure via simplified beam theory. In order to solve the mechanical model, the shooting method, a numerical method to solve ordinary differential equation (ODE) is employed. Finite element analyses (FEA) are also performed to verify the developed theoretical model. The influences of the geometric parameters on deformations and forces are analyzed to achieve the optimal design of the structures. The stretching tests of horseshoe-shaped samples manufactured by three-dimensional (3D) printing are implemented, whose results shows a good agreement with those from theoretical predictions. The developed models can serve as the guidelines for the design of flexible solar arrays in spacecraft.

## 1. Introduction

Stretchable electronic devices [1] have exploited wide applications in the fields of aerospace [2,3,4,5], biomedicine [6,7], intelligent wear [8], etc., which helps to tackle many challenges in engineering and becomes the research focus at present. There are two approaches to achieve the stretchability of electronic devices. One is to directly employ the intrinsically stretchable organic materials [9,10], and the other one is to manipulate the structures with large deformation and small strain through geometric engineering [11,12,13]. The latter is more easily to enjoy the development of modern electronic technology. Through reasonable structural design, electronic devices can be guaranteed to withstand smaller strain under large deformation to protect themselves. In order to solve the above two seemingly contradictory requirements, multifarious solutions have been proposed such as wavy structural configuration [14,15], island-interconnect configuration [16,17], fractal design of stretchable interconnects [18,19] and origami and kirigami structural configurations [20,21,22]. Among them, as a fresh structural design method, the origami structures can achieve more complex and novel structures that cannot be accomplished by other methods [23,24,25,26].

When turn the gaze to the solar arrays of spacecraft, one can find the similar requirements as in stretchable electronics. Solar cells are characterized by longevity, high efficiency, small volume, light weight and large spread area. The traditional inorganic solar cells such as silicon and gallium arsenide solar cells can meet the above characteristics. However, the material brittleness becomes the first safety and reliability concerns during their in-orbit operations. Inspired by flexible electronics, a flexible substrate design seems to be a good solution [27]. The traditional substrates of solar cell array include rigid substrate, semi-rigid substrate and flexible substrate. In recent years, the new types of structure substrates, including the origami/kirigami structures [22,28], are proposed.

In this paper, a new origami design is proposed. The horseshoe-shaped design is introduced as shown in Figure 1a. The structure can not only achieve large deformation, but also guarantee small strain at the particular region. This design can also manipulate the parts of large and small deformations during stretching, which can prevent the damage of the core devices. The finite deformation beam theory [16] is employed to establish an analytical model of the horseshoe-shaped structure, which can be extended to the analyses of more complex structures. The governing equations obtained by the finite deformation theory are solved by shooting method, a numerical solution to solve boundary value problems for ordinary differential equations (ODE). It transforms the boundary value problem of ODE into an initial value problem. The finite element analyses (FEA) are performed to verify the developed theoretical model. The comparisons of the deformation and tension force from theoretical analysis and FEA validate the correctness of our theoretical model. Then the influence of various geometrical parameters of horseshoe-shaped structures on deformation and tension force are investigated, which provides a basis for the adjustment of deformation and force. Finally, the uniaxial tensile tests of the horseshoe-shaped structures prepared by 3D printing are performed, whose results shows great coincidence with those from theoretical model. The mechanical behaviors of horseshoe-shaped structures are studied by theoretical analysis, finite element analysis and experiment, which showed a high degree of consistency.

## 2. Analytical Model

For slender curved beam structures with the thickness-to-length ratio smaller than 0.05, the Euler beam theory is usually adopted to model the deformations, where the effects of axial elongation and shear strain are neglected. According to periodicity and symmetry, the mechanical model can be simplified into a three-section curved beam structure as shown in Figure 1b. It contains two straight parts (I and II) denoted by *L*_1_ and *L*_3_, respectively, and an arc part (III) with a radio *R* and central angle *α* in the middle. The applied displacement load is denoted as *u*_app_ on one unit period structure. The real displacement load on a quarter period is *u*_app_/4 when the right endpoint is fixed. As shown in the Figure 1c, the representative infinitesimal arc section *AB* is built in initial coordinate *S*, which represents the state before deformation. While, the arc *ab* is built in current coordinate *s*, which denotes the state after deformation. Now take a point *P* (*X*, *Y)* on the infinitesimal arc section *AB*, and its tangent angle is *Θ*. After deformation, it becomes point *p* (*x*, *y)*, and the tangent angle is *θ*. The equilibrium differential equations represented in the current coordinate system can be obtained by force equilibriums of the deformed infinitesimal arc section *ab*:(1)$$\left\{ \begin{array}{l} \frac{dn}{ds}-q\frac{d\theta }{ds}=0\\ \frac{dq}{ds}+n\frac{d\theta }{ds}=0\\ \frac{dm}{ds}=q \end{array}\right.$$
where, *n* represents axial force. *q* denotes shear force. *m* is bending moment in the current configuration.

The following assumptions are introduced: the curved beam of horseshoe-shaped structure keeps its original length after deformation, i.e.,
(2)$$\lambda =\frac{ds}{dS}=1$$
where *λ* represents the elongation of the curved beam.

The physical equations can be written as
(3)$$\left\{ \begin{array}{l} n=EA\left(\lambda -1\right)\\ m=EI\Delta \kappa \end{array}\right.$$
where, *E* is Young’s modulus. *EA* and *EI* denote tensile stiffness and bending stiffness, respectively. Δ*κ* represents the change of curvature. Κ and *κ* denote the curvature of curved beams before and after deformation, respectively. Then, Δ*κ* can be expressed as
(4)$$\Delta \kappa =\kappa -Κ=-\frac{d\theta }{ds}-Κ=-\frac{d\theta }{dS}-Κ$$

Based on the Equation (2), the coordinate transform relationship can be written as
(5)$$\left\{ \begin{array}{l} \frac{dx}{dS}=\cos\theta \\ \frac{dy}{dS}=\sin\theta \end{array}\right.$$

According to the first two lines in equilibriums Equation (1), the differential equations of shear force and axial force can be obtained respectively. Combined with the boundary conditions, the specific differential equations are obtained as follows
(6)$$\left\{ \begin{array}{l} \frac{d^{2}q}{d\theta^{2}}+q=0\\ q|{}_{\theta =\theta_{0}}=q_{0}\\ \left(-\frac{dq}{d\theta }\right)|{}_{\theta =\theta_{0}}=n_{0}, \end{array}\right.\;\;\left\{ \begin{array}{l} \frac{d^{2}n}{d\theta^{2}}+n=0\\ n|{}_{\theta =\theta_{0}}=n_{0}\\ \frac{dn}{d\theta }|{}_{\theta =\theta_{0}}=q_{0} \end{array}\right.$$
where, the subscript “0” represents the initial endpoint of each beam. The general solutions to Equation (6) are
(7)$$\left\{ \begin{array}{l} q=q_{0}\cos\left(\theta -\theta_{0}\right)-n_{0}\sin\left(\theta -\theta_{0}\right)\\ n=n_{0}\cos\left(\theta -\theta_{0}\right)+q_{0}\sin\left(\theta -\theta_{0}\right) \end{array}\right.$$

The tension and shear force can also be written directly according to the equilibrium conditions. The third equation in equilibrium Equation (1), the second equation in physical Equation (3) and curvature change Equation (4) yield to
(8)$$\frac{d^{2}\theta }{dS^{2}}=-\frac{q}{EI}$$

Substituting the expression for *q* in Equation (7) into the Equation (8), one obtains
(9)$$\frac{d^{2}\theta }{dS^{2}}=-\frac{1}{EI}\left[q_{0}\cos\left(\theta -\theta_{0}\right)-n_{0}\sin\left(\theta -\theta_{0}\right)\right]$$

After one time of length integration, one obtains
(10)$$\frac{d\theta }{dS}=sign\left(\frac{d\theta }{dS}\right)\sqrt{C\left[G-\cos\left(\theta -B\right)\right]}$$

The above equation is so-called the governing equation, where the plus or minus is determined by the sign of the moment at that point. *B*, *C* and *G* are constants:$$\left\{ \begin{array}{l} B=\varphi +\theta_{0}\\ \varphi =\arctan\frac{q_{0}}{n_{0}}\\ C=\frac{2\sqrt{n_{0}^{2}+q_{0}^{2}}}{EI}\\ G=\frac{EI}{2\sqrt{n_{0}^{2}+q_{0}^{2}}}{\left(\frac{m_{0}}{EI}+Κ\right)}^{2}+\cos\varphi \end{array}\right.$$

Here, we introduce a function,
(11)$$F\left(\theta \right)=\sqrt{C\left[G-\cos\left(\theta -B\right)\right]}$$

Then the length of the curved beam can be obtained by Integration of Equation (10)
(12)$$S=\int_{\theta_{0}}^{\theta }sign\left(\frac{d\theta }{dS}\right)\frac{d\theta }{F\left(\theta \right)}$$

The coordinates of any point in curved beams can also be found by
(13)$$\left\{\begin{array}{c} x\\ y \end{array}\right\}=\left\{\begin{array}{c} x_{0}\\ y_{0} \end{array}\right\}-\int_{\theta_{0}}^{\theta }sign\left(\frac{d\theta }{dS}\right)\frac{1}{F\left(\theta \right)}\left\{\begin{array}{c} \cos\theta \\ \sin\theta \end{array}\right\}d\theta$$

The forces of the curved beam can also be easily expressed as
(14)$$\left\{ \begin{array}{l} n=n_{0}\cos\left(\theta -\theta_{0}\right)+q_{0}\sin\left(\theta -\theta_{0}\right)\\ q=q_{0}\cos\left(\theta -\theta_{0}\right)-n_{0}\sin\left(\theta -\theta_{0}\right)\\ m=m_{0}+\left(y_{0}-y\right)\left(n_{0}\cos\theta_{0}-q_{0}\sin\theta_{0}\right)\\ \;\;\;-\left(x_{0}-x\right)\left(n_{0}\sin\theta_{0}+q_{0}\cos\theta_{0}\right) \end{array}\right.$$

The two connected beams are coupled together by continuity conditions
(15)$$\left\{ \begin{array}{l} \theta_{21}=\theta_{12},x_{21}=x_{12},y_{21}=y_{12}\\ \theta_{31}=\theta_{22},x_{31}=x_{22},y_{31}=y_{22} \end{array}\right.$$
where the first subscript indicates the number of the beam, and the second one indicates the number of the endpoint along *s* coordinate. The boundary conditions are derived from symmetry conditions of the beams structure
(16)$$\left\{ \begin{array}{l} \theta_{11}=0\\ q_{11}=0\\ x_{32}=X_{32}\\ y_{32}=Y_{32}\\ \theta_{32}=\alpha \end{array}\right.$$

A supplementary condition can be provided by the expression of bending moment
(17)$$m_{11}=n_{11}\left[R\left(1-\cos\alpha \right)+\frac{L_{3}}{2}\sin\alpha -v_{11}\right]$$

If displacement load *u*_11_ is given, the unknown independent parameters are *n*_11_ and *v*_11_ in the initial point. The next section is to find a method to solve the above two parameters.

## 3. Solution Method

The boundary value problem of ODE in Equation (10) cannot be solved analytically, so an effective numerical solution is needed. Moreover, the presented structure is composed of three curved beams, and the governing equation is only applicable to a single part of the structure. Actually, three coupled differential equations rather than one should be dealt with. In this paper, the shooting method is employed, which can convert the boundary value problem into an initial value problem of ODE.

According to the symmetry condition of the structure, *θ*_11_ = 0, and *q*_11_ = 0 are set at the initial endpoint of beam I. According to the anti-symmetry condition of the structure, *u*_32_ = *v*_32_ = 0 and *θ*_32_ = *α* are assigned at the end point of beam III. A vector *P_ij_* is used to represent the displacements and forces components of the *j*th end of *i*th beams as follows
(18)$$P_{ij}=\left\{u_{ij},v_{ij},\theta_{ij},n_{ij},q_{ij},m_{ij}\right\}\;\;\left(i=1,2,3;\;j=1,2\right)$$

When the initial values of *n*_11_ and *v*_11_ are given, *P*_12_ can be obtained by Equations (12)–(14). Then, according to continuity conditions (Equation (15)), *P*_21_ = *P*_12_ is obviously. Going through the same process twice, one can obtain the *P*_32_. Then, the obtained results are compared with the boundary conditions in Equation (16). If the differences are within the allowable error, the solution is found. Otherwise, adjust the initial values according to Newton iteration method and recalculate again. Generally, one or two iterations can make the results converge closely to the true value due to its high rate of convergence. Two or more iterations can make the error percentage achieve 0.001%. The flow diagram is showed in Figure 2. The mathematical software MAPLE undertakes the entire calculation process.

## 4. Finite Element Analyses

In this section, three-dimensional finite element analyses (FEA) were employed to validate the above analytic model. In the FEA, eight-node linear brick, reduced integration, hourglass control solid elements were adopted to model the horseshoe-shaped structure. The geometries are shown in Figure 1a. The specific mechanical and geometric parameters are shown in Table 1, which is consisting with those in the following experiment section. Specific displacement loadings are applied at both ends of the structure.

In Figure 3, the applied strain *ε_app_* is defined as the ratio of the end point displacement to the overall original length of the structure, i.e.,
(19)$$\varepsilon_{app}=\frac{u_{11}}{L_{1}+\left(L_{3}-R\right)\cos\alpha }\times 100\%$$

In Figure 3, the configurations of the curved beam predicted by the analytical model fix well with those from FEA. The variation of the applied axial forces on the end with the applied strain is shown in the Figure 4a, which also show good consistency. Figure 4b shows the variation of strain on the upper surface of a half-periodic horseshoe-shaped structure (shown in the inside of Figure 4b) with respect to 30% applied strain. Because of the neglecting of shear strain, the longitudinal strain predicted by theoretical model has some deviation. However, the first concern is always the deformation of the substrate, which determine the strain in the upper mounted solar cells. The maximum strain occurs at arc beam section where the solar cells are not placed.

In order to optimize the structure design, it is necessary to study the influences of geometric parameters of curved beams. The objective of optimization is to minimize the strain on the upper surface of the substrate. There are five geometric parameters, *L*_1_, *L*_3_, *α*, *R* and *h*. After dimensionless, the independent parameters are *α*, *R*/*L*_1_, *L*_3_/*L*_1_, h/*L*_1_. The basic parameters are set as *L*_1_ = 20 mm, *L*_3_ = 20 mm, *R* = 4 mm, *α* = $\frac{5\pi }{6}$, *h* = 2 mm and *b* = 20 mm.

Figure 5 shows the effect of the four dimensionless parameters on strain and force at the both ends. The left ordinate represents the maximum strain on the upper surface of the structure and the right ordinate represents the force applied on both ends of the horseshoe-shaped structure. It can be seen from Figure 5a that strain and force both decrease first and then increase with the increase of *α*, and the minimum value appears near $\frac{5\pi }{6}$. Figure 5b indicates the strain and force decrease gradually with the arc radius. However, the size effects on fabrication and assembly should also be considered in design. Figure 5c shows the influence of *L*_3_, which is the same as the influence of radius, and also shows a decreasing trend with increasing *L*_3_. The influence of thickness is shown in Figure 5d. The strain increases linearly with the increase of thickness, and the increase rate of force follows an approximate exponential growth. Therefore, based on the above factors and the actual situation, the combinations of the central angle of $\frac{5\pi }{6}$, a larger radius *R*, a larger *L*_3_ and a smaller *h*, shows a better strain optimization of the structures.

## 5. Experiments

This section further verifies the correctness of theoretical analyses via tensile experiments. The experiments aim to obtain the deformations and the applied forces of the horseshoe-shaped structures during stretching within the linear elastic range of the material, and verify the accuracy of the theory in previous sections. Here, the horseshoe-shaped structure is prepared by 3D printing technology. As shown in Figure 6, the experiments are carried by INSTRON 5900 (measuring range 500N, accuracy 0.4%, INSTRON Co., Norwood, MA, USA). The deformations of the structures are captured by digital camera.

The selected material is thermoplastic polylactic acid (PLA). The specific geometric dimensions are shown in Table 1, which is consisting with the configurations in Figure 1a. A clamping part with a length of 20 mm is reserved at both ends. At the same time, the same printing parameters are used to fibrate the standard tensile samples, which are used for elastic modulus and linear elastic range testing. In the tensile experiment, a stretcher machine with a measuring range of 500 N is used, as shown in Figure 6b, and the tensile rate is 2 mm/min. During the experiment, deformation maps under different tensile displacements are obtained by digital camera shooting. In order to reduce errors, the digital camera was fixed by triangular bracket, and the angle of view was adjusted to make the camera directly facing the target. The deformation diagram is shown in Figure 6c, where an applied deformation of 10%, 20%, 30% and 40% are adopted, respectively.

The elastic modulus and elastic range are 2.5 GPa and 1.2%, respectively, obtained by standard tensile test. In order to reduce the error, the middle period of the horseshoe-shaped structure in Figure 6c was selected for comparisons. The comparison results are shown in Figure 7a, where the overall deformations of 10%, 20%, 30% and 40% are applied, respectively. In Figure 7a, the deformed configurations of the structures from experiment and analytical model fix well for the applied deformation from 0 till 40%. The relations between axil force and applied deformation also show high consistencies among results from experiment, analytical model and FEA, as shown in Figure 7b. It shows that within the range of linear elasticity of material, this proposed theoretical method is a powerful tool in design of horseshoe-shaped structures.

Several solar array samples are fabricated to further illustrate the potentials of the proposed horseshoe-shaped structures in solar arrays applications. The flexible substrate is made of polydimethylsiloxane (PDMS). Nine foursquare solar cells with edge length of 10 mm and thickness of 0.15 mm are adhered to polyethylene terephthalate (PET) shell structures, which are bonded to flexible substrate to maintain origami configurations (Figure 8a). The solar arrays can bear 40% stretching (Figure 8b) and can conformably deform to spherical (Figure 8c) or cylindrical (Figure 8d) surfaces, respectively. In experiments, there is no adhesive failure or solar cell damage after hundreds times of stretching. In addition, the solar array structures recover to their initial state after standing for a few minutes.

## 6. Conclusions

In this paper, we propose a new origami substrate of solar arrays. A simplified theoretical model is established via finite deformation theory for curved beam. The governing equations for curved beam are derived and solved by shooting method. The results are in good agreement with both of those from FEA and experiments. The theoretical analysis shows a high accuracy in predictions of deformations and forces of the horseshoe-shaped structures. This theory may serve as the guidelines for design of flexible solar arrays in engineering applications.

## Figures and Tables

**Figure 1 micromachines-13-00732-f001:**
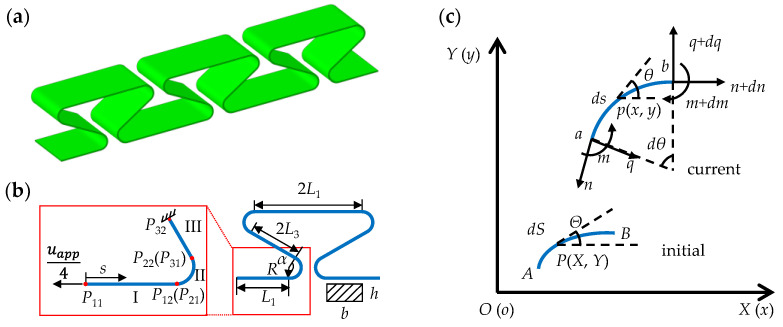
The diagram of horseshoe-shaped structure. (**a**) The three-dimensional view of horseshoe-shaped structure with three periods; (**b**) the simplified mechanical model; (**c**) the infinitesimal arc sections before and after deformation.

**Figure 2 micromachines-13-00732-f002:**
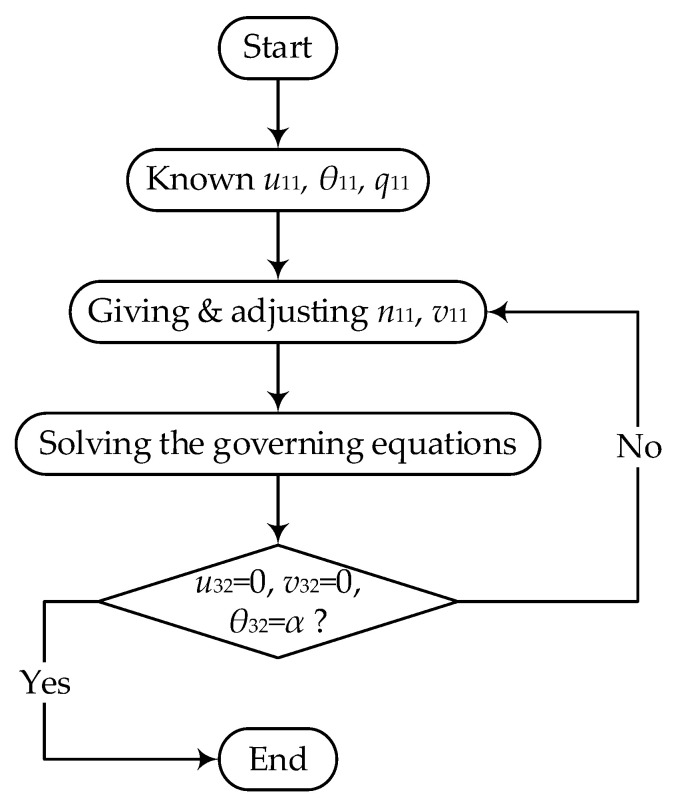
Solution flow chart of the shooting method.

**Figure 3 micromachines-13-00732-f003:**
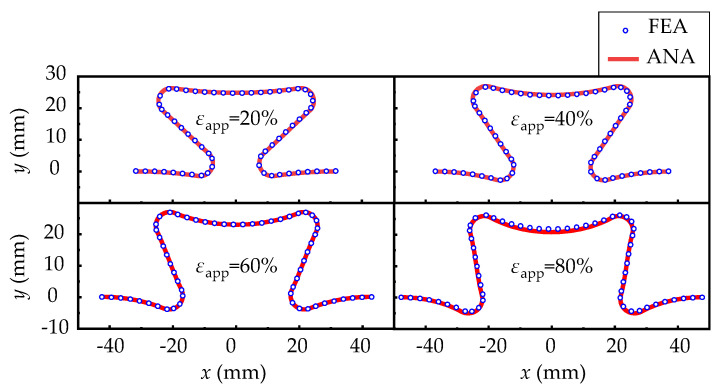
The deformations between FEA and analytic results when applied strains are 20%, 40%, 60% and 80%, respectively.

**Figure 4 micromachines-13-00732-f004:**
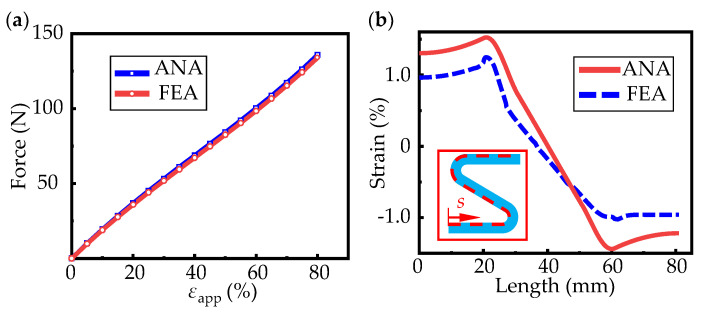
The comparisons between analytical and FEA results. (**a**) The variation of force with ap-plied strain; (**b**) the variation of strain along the path showed inside of figure.

**Figure 5 micromachines-13-00732-f005:**
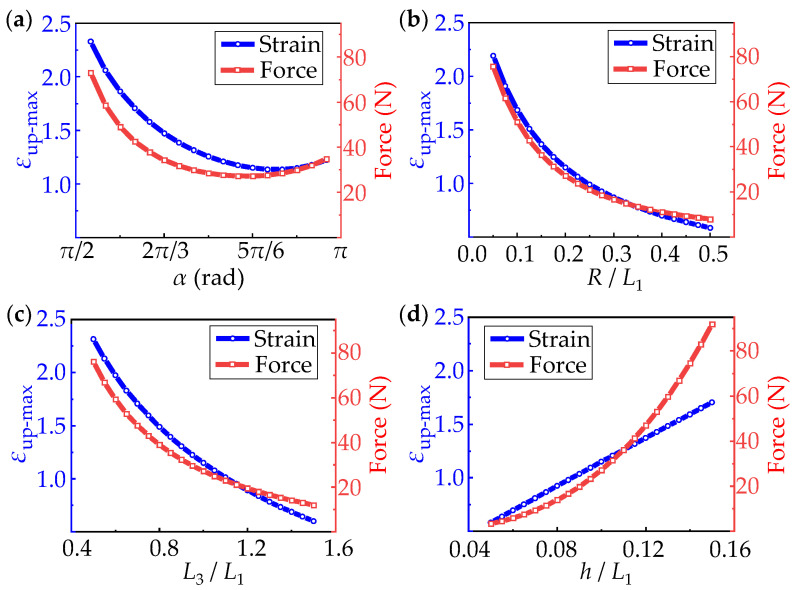
The influences of non-dimensional parameters (**a**) *α*; (**b**) *R*/*L*_1_; (**c**) *L*_3_/*L*_1_; (**d**) h/*L*_1_ on strain and force of horseshoe-shaped structures during deformation.

**Figure 6 micromachines-13-00732-f006:**
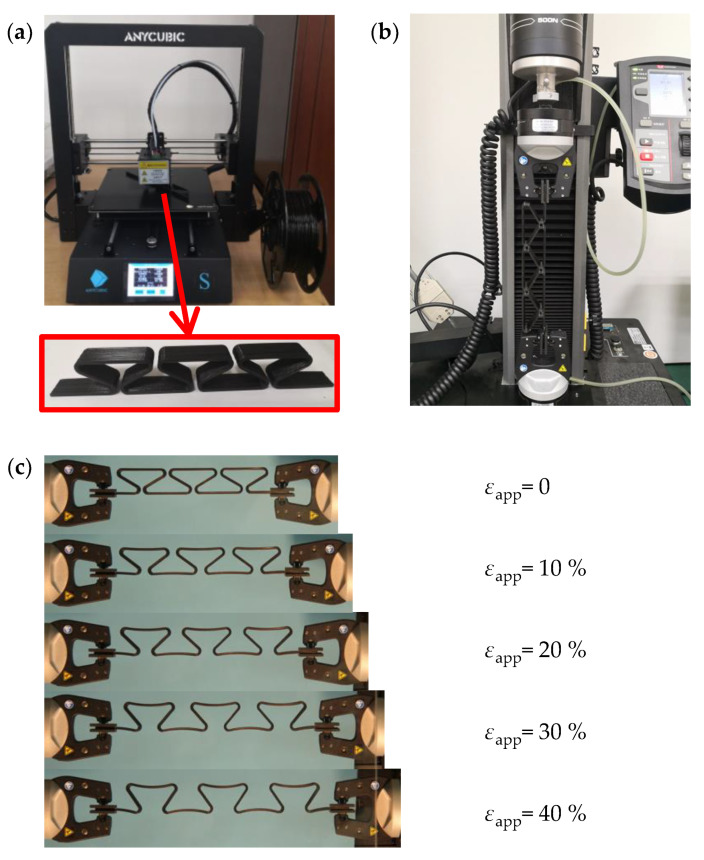
The experiments of the horseshoe-shaped structure. (**a**) The horseshoe-shaped structure prepared by 3D printer; (**b**) The stretching test through stretcher machine; (**c**) The stretching test results obtained by digital camera.

**Figure 7 micromachines-13-00732-f007:**
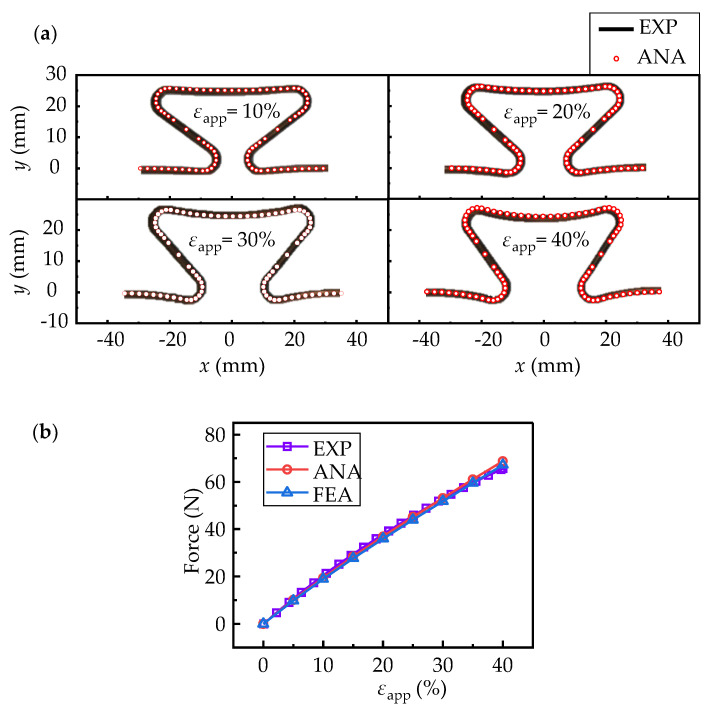
The comparisons between experiments and analytical results: (**a**) deformed configurations with applied deformations of 10%, 20%, 30% and 40%, respectively; (**b**) force variations with applied deformations. The square, round and triangle signs represent results from experiment, analytical model and FEA, respectively.

**Figure 8 micromachines-13-00732-f008:**
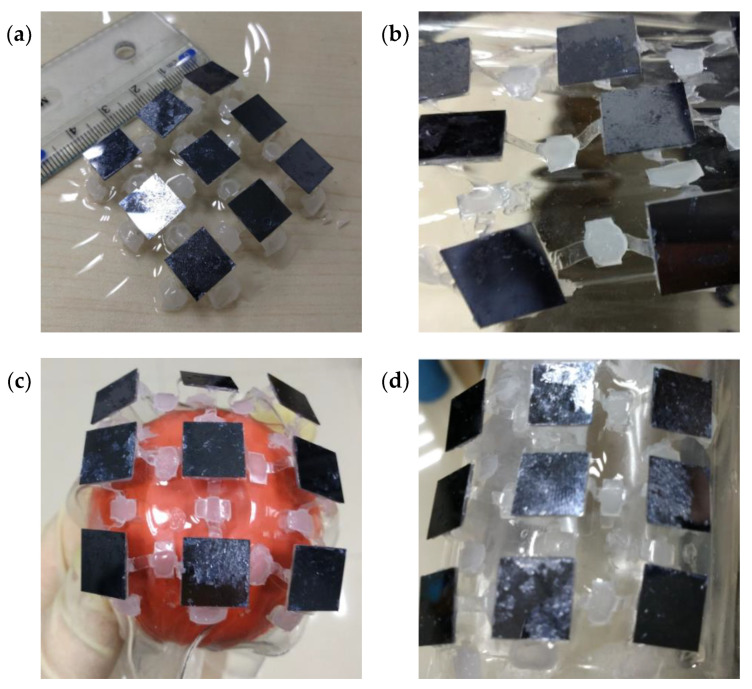
Experiments about solar array structures: (**a**) the initial structure; (**b**) stretching to 40%; deform conformably to (**c**) spherical and (**d**) cylindrical surfaces, respectively.

**Table 1 micromachines-13-00732-t001:** Geometrical parameters and mechanical parameters.

*L*_1_/mm	*R*/mm	*α*	*L*_3_/mm	*h*/mm	*b*/mm	*E*/GPa
20	4	$\frac{5\pi }{6}$	20	2.5	20	2.5

## Data Availability

Not applicable.

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
