# Peer review of "Mechanical Behaviors of the Origami-Inspired Horseshoe-Shaped Solar Arrays"

_micromachines, 2022, doi:10.3390/mi13050732_

Round 1

Reviewer 1 Report

This work demonstrates the calculations of horseshoe-shaped stretchable structures and proposes their applications in solar arrays. The stretchability is one of the demands from space solar panels and this manuscript supplies a simplified design approach.

The manuscript is well organized. The theoretical model is well validated by the FEA and experiments, respectively. I would like to recommend this manuscript for publication pending the following minor revisions.

  1. The horseshoe-shaped structure has been well studied by many scholars. Some important work should be reviewed in the introduction part, e.g., “Postbuckling analysis and its application to stretchable electronics. Journal of the Mechanics and Physics of Solids, 2012, 60(3), 487-508”.

  1. In Part 4 FEA, “The specific mechanical and geometric parameters are shown in Table 1, which is consisting with the those in the following experiment section.”, “the” is redundant.

  1. For the calculation, the general number of iterations or calculation time should be demonstrated.

Author Response

Comment 1. The horseshoe-shaped structure has been well studied by many scholars. Some important work should be reviewed in the introduction part, e.g., “Postbuckling analysis and its application to stretchable electronics. Journal of the Mechanics and Physics of Solids, 2012, 60(3), 487-508”.

Our response: Thanks for the comments. We have added the comments for the relevant references in the manuscript.  

Comment 2.      In Part 4 FEA, “The specific mechanical and geometric parameters are shown in Table 1, which is consisting with the those in the following experiment section.”, “the” is redundant.

Our response: Thank the reviewer for pointing this typo. We have deleted the redundant “the” in main text.

Comment 3. For the calculation, the general number of iterations or calculation time should be demonstrated.

Our response: We thank the reviewer for pointing out this key issue. The calculation time depends on the number of iterations and computer performance, so we add the content of the general number of iterations to the main text as following.

Modifications: In Page 11, we have added “Generally, one or two iterations can make the results converge closely to the true value due to its high rate of convergence. Two or more iterations can make the error percentage achieve 0.001%.”

Reviewer 2 Report

This work demonstrated a horseshoe-shaped substrate, which can combine rigid solar cells and soft substrates, and prevent damage during deformations. Moreover, the stretching tests of horseshoe-shaped samples manufactured by three-dimensional printing are implemented, whose results show a good agreement with theoretical predictions. In this regard, I recommend the publication of this work in micromachines. Some detailed comments are:

1. Since the authors emphasize the application of this substrate in solar cells, would the authors provide any data on their solar cells? If not, the authors do not need to emphasize the application of this substrate in solar cells.

Author Response

Comment 1. Since the authors emphasize the application of this substrate in solar cells, would the authors provide any data on their solar cells? If not, the authors do not need to emphasize the application of this substrate in solar cells.

Our response: We thank the reviewer for pointing out this key issue. We have added experiments results about solar array structures based the theoretical analyses in Part 5.

Modifications: In Page, we added Fig. 8 and relevant comments. “Several solar array samples are fabricated to further illustrate the potentials of the proposed horseshoe-shaped structures in solar arrays applications. The flexible substrate is made of polydimethylsiloxane (PDMS). Nine foursquare solar cells with edge length of 10 mm and thickness of 0.15 mm are adhered to polyethylene terephthalate (PET) shell structures, which are bonded to flexible substrate to maintain origami configurations (Figure 8a). The solar arrays can bear 40% stretching (Figure 8b) and can conformably deform to spherical (Figure 8c) or cylindrical (Figure 8d) surfaces, respectively. In experiments, there is no adhesive failure or solar cell damage are hundreds times of stretching. In addition, the solar array structures recover to their initial state after standing for a few minutes.”